# Characterization of an *Mtbp* Hypomorphic Allele in a Diethylnitrosamine-Induced Liver Carcinogenesis Model

**DOI:** 10.3390/cancers15184596

**Published:** 2023-09-16

**Authors:** Atul Ranjan, Elizabeth A. Thoenen, Atsushi Kaida, Stephanie Wood, Terry Van Dyke, Tomoo Iwakuma

**Affiliations:** 1Department of Pediatrics, Children’s Mercy Research Institute, Kansas City, MO 64108, USA; 2Department of Cancer Biology, University of Kansas Medical Center, Kansas City, KS 66160, USA; 3Department of Pathology & Laboratory Medicine, University of Kansas Medical Center, Kansas City, KS 66160, USA; 4Path Forward Solutions, LLC, Frederick, MD 21701, USA

**Keywords:** MDM2 binding protein, MTBP, hypomorphic allele, mouse, liver cancer, migration

## Abstract

**Simple Summary:**

The MDM2 binding protein MTBP is implicated in various cellular functions and cancer-related processes, which vary depending on the cellular context and its localization within the cell. Moreover, the in vivo physiological function of MTBP remains unclear. To overcome embryonic lethality due to complete deletion of the *Mtbp* gene in mice, we created mice with an *Mtbp* hypomorphic allele (*Mtbp^H^*) that expresses Mtbp protein at approximately 30% of the wild-type level. In a carcinogen-induced liver cancer model, *Mtbp^H/−^* mice showed worse overall survival than wild-type mice. MEFs generated from the *Mtbp^H/−^* mice displayed an increased nuclear localization of p-Erk1/2 protein and enhanced migratory potential. Thus, the newly generated *Mtbp^H/−^* mice and MEFs can be used to study the in vivo physiological function of Mtbp and validate its diverse functions observed in human cells.

**Abstract:**

MTBP is implicated in cell cycle progression, DNA replication, and cancer metastasis. However, the function of MTBP remains enigmatic and is dependent on cellular contexts and its cellular localization. To understand the in vivo physiological role of MTBP, it is important to generate *Mtbp* knockout mice. However, complete deletion of the *Mtbp* gene in mice results in early embryonic lethality, while its heterozygous deletion shows modest biological phenotypes, including enhanced cancer metastasis. To overcome this and better characterize the in vivo physiological function of MTBP, we, for the first time, generated mice that carry an *Mtbp* hypomorphic allele (*Mtbp^H^*) in which Mtbp protein is expressed at approximately 30% of that in the wild-type allele. We treated wild-type, *Mtbp^+/−^*, and *Mtbp^H/−^* mice with a liver carcinogen, diethylnitrosamine (DEN), and found that the *Mtbp^H/−^* mice showed worse overall survival when compared to the wild-type mice. Consistent with previous reports using human liver cancer cells, mouse embryonic fibroblasts (MEFs) from the *Mtbp^H/−^* mice showed an increase in the nuclear localization of p-Erk1/2 and migratory potential. Thus, *Mtbp^H/−^* mice and cells from *Mtbp^H/−^* mice are valuable to understand the in vivo physiological role of Mtbp and validate the diverse functions of MTBP that have been observed in human cells.

## 1. Introduction

MDM2 binding protein (MTBP) was originally identified using the yeast two-hybrid system as a protein capable of binding to MDM2 [1]. Abundant literature indicates that the biological functions of MTBP are independent of MDM2, and MTBP’s function appears to be dependent on cellular contexts or the intracellular location of MTBP [2,3,4,5,6]. Until today, diverse functions of MTBP have been reported. These include mitotic progression, DNA replication, proliferation, cell migration, and cancer metastasis [2,5,7,8,9,10,11,12]. Specifically, our previous studies and others have demonstrated that MTBP suppresses migration and metastasis of osteosarcoma, gastric cancer, head and neck cancer, and hepatocellular carcinoma (HCC) cells [2,9,10,11,12,13].

The first observation of the migratory- and metastasis-suppressive role of MTBP was made using heterozygous *Mtbp* knockout mice in which tumors from *Mtbp^+/−^p53^+/−^* mice showed a higher frequency of metastasis of osteosarcoma and HCC (~20%) when compared with tumors from *p53^+/−^* mice (~3%) [10]. Clinical studies also support that decreased MTBP levels are correlated with increased metastasis and/or poor prognosis in gastric cancer, head and neck cancer, esophageal squamous cell carcinoma, and HCC [9,11,13,14]. Although the underlying mechanisms need to be clarified, the involvement of actinin-4 (ACTN4, an actin-bundling protein involved in cell motility and migration) for osteosarcoma and lung cancer as well as Erk1/2 for HCC has been shown [2,11,12,15]. Intriguingly, the C-terminal region of MTBP is required for its inhibitory binding to ACTN4 and Importin 7, a protein that shuttles phosphorylated Erk1/2 into the nucleus [2,11,12]. Such migration-suppressive function of MTBP may be mediated by the cytoplasmic portion of MTBP, since a nuclear localization signal (NLS) mutant MTBP, localizing to the cytoplasm, retains the ability to inhibit cancer cell migration [11,12]. These results suggest that cytoplasmic MTBP may function as a migration/metastasis suppressor, while nuclear MTBP may play roles in the regulation of mitotic progression and DNA replication [5,7]. Additionally, there are reports that show oncogenic roles of MTBP where MTBP overexpression enhances cell proliferation, transformation, and migration through interactions with Myc and Ets-1 [4,16,17,18,19]. Moreover, Grieb et al. [20] recently reported that *Mtbp* heterozygous mice showed an increase in longevity (a 20% increase in median survival) and levels of metabolic markers in the liver as compared to wild-type mice. Thus, physiological and pathological functions of MTBP may vary depending on cellular contexts and intracellular localization of MTBP.

To understand the in vivo function of MTBP and validate the diverse and sometimes controversial observations in MTBP functions, it is essential to generate and analyze *Mtbp* knockout mice. However, early embryonic lethality following homozygous deletion of *Mtbp* in mice made it impossible to analyze the phenotypes induced by complete Mtbp deletion [10]. Indeed, all in vivo studies have been made using *Mtbp*^+/−^ mice, while the phenotypes observed in *Mtbp*^+/−^ mice are modest [10]. In order to overcome these existing challenges, we generated mice carrying a low-level expression (hypomorphic) allele for *Mtbp* using a *hygromycin* cassette [21]. This approach has successfully been used to uncover the in vivo phenotypes of several other genes, including *Bub1*, *Bub3*, and *BubR1*, whose complete deletion causes embryonic lethality, while the deletion of one allele shows unnoticeable or modest phenotypes, similar to Mtbp [22,23,24]. Since mRNA and protein levels of MTBP in human HCC tissues are reduced to 30% compared to those in adjacent non-tumor liver tissues [11], we analyzed *Mtbp* hypomorphic (*Mtbp^H^*) mice in a carcinogen-induced liver carcinogenesis model. In addition, we generated and analyzed mouse embryonic fibroblasts (MEFs) derived from *Mtbp^H^*^/*−*^ mice for their migratory potential.

## 2. Materials and Methods

### 2.1. Generation of Mtbp Hypomorphic Mice

To generate the *Mtbp* hypomorphic (*Mtbp^H^*) allele, we obtained the *hyg* cassette (TKhygpA) from Dr. Wieringa [21,25]. This cassette was modified to be flanked by *loxP* and *Frt* sequences, which was inserted into the *Mtbp* intron 5. We also inserted another loxP site into intron 6 to have the option to generate a conditional *Mtbp* knockout allele. Successfully targeted 129Sv/Ev ES clones were injected into blastocysts in the National Cancer Institute (NCI) genetically engineered mouse facility. The resulting heterozygous (*Mtbp^H/+^*) mice were backcrossed to C57BL/6 mice for 8 generations so that they had over 99% of the C57BL/6 background.

Additionally, we generated *Mtbp* knockout (*Mtbp^−^*) mice by inserting the *pGKneo* cassette with the *SV40 poly(A)* sequence in the same direction as the *Mtbp* gene (Appendix A). Although there was a *loxP* site in intron 7 in the targeting vector, this site was not inserted into the ES cell genome. The homozygous mice turned out to be early embryonic lethal, consistent with a previous report [10], confirming that this allele is a null allele (*Mtbp^−^*). All mouse studies were conducted in compliance with Institutional Animal Care and Use Committee protocols of the University of Kansas Medical Center (KUMC).

### 2.2. Genotyping

Genomic DNA was isolated from the tails, which were incubated in tail lysis buffer containing 0.2 mg/mL of proteinase K at 55 °C overnight, using the standard genomic DNA isolation protocol. Mouse genotyping for the hypomorphic *Mtbp^H^* allele was performed with PCR amplification using primer sets of I6F (5′-cac agg act tac cat gtc ctg tct gt-3′) and E7R (5′-ata tcc aga gtt gtc acc cct acg gt-3′), while the *Mtbp*^−^ allele was detected using primers of NeoSVF (5’-gaa ttc gcc ctt cga cta gcc ata atc agc-3′), I5F (5′-cta gct atg ctg gag aat tag caa gc-3′), and E6R (5’-gag ggt ctt tgt cag aag gca aca g-3′). The PCR products were resolved on 1.5% agarose gels.

### 2.3. DEN-Induced Liver Carcinogenesis

Diethylnitrosamine (DEN, N0258, Sigma-Aldrich, St. Louis, MO, USA,) was injected once intraperitoneally to 2-week-old wild-type (*Mtbp^+/+^*), *Mtbp^+/−^*, and *Mtbp^H/−^* mice (25 mg/kg body weight). The mice were euthanized when they became moribund.

### 2.4. Antibodies

The antibodies used for western blotting included mouse monoclonal anti-Mtbp (sc-137201, Santa Cruz, CA, USA) and rabbit polyclonal anti-Gapdh (sc-27117, Santa Cruz, CA, USA) antibodies. For the immunofluorescence studies, rabbit monoclonal anti-p-Erk1/2 Thr202/Tyr204 (#4370, Cell Signaling, Danvers, MA, USA) antibody was used. For IHC, rabbit monoclonal anti-p-Erk1/2 Thr202/Tyr204 (#4370, Cell Signaling) and goat polyclonal anti-Mtbp (sc-47174, Santa Cruz, CA, USA) antibodies were used.

### 2.5. Immunofluorescence

The cells plated onto poly-D-lysine/laminin-coated glass coverslips (BD Biosciences) were incubated with 4% paraformaldehyde for 20 min. After blocking with 1% BSA in PBS with 0.1% Tween 20 (PBS-T), the cells were incubated with primary antibodies overnight at 4 °C and subsequently with the appropriate secondary antibodies. The samples were mounted in the ProLong Gold Antifade Reagent with DAPI (Invitrogen, Waltham, MA, USA). The results were analyzed using a Nikon epifluorescence microscope (Nikon, Tokyo, Japan).

### 2.6. Quantitative Reverse Transcriptase PCR (qRT-PCR)

The total RNA from the mouse cells isolated using *Quick*-RNA Miniprep Kit (Zymo Research, Irvine, CA, USA) was reverse-transcribed with the M-MLV Reverse Transcriptase Kit (Invitrogen, Waltham, MA, USA) using 1 µg total RNA from each sample. The mouse mRNA expression for *Mtbp* and *Gapdh* was analyzed with quantitative RT-PCR (qRT-PCR) with TaqMan probes for *Mtbp (Mm00519571_m1_g1*, Thermo Fisher Scientific, Waltham, MA, USA) and *Gapdh (Mm99999915_g1*) using Applied Biosystems ViiA7 (Life Technologies, Carlsbad, CA, USA). *Mtbp* mRNA levels were normalized to those of *Gapdh* mRNA.

### 2.7. 3T3 Assay

MEFs were prepared from 13.5-day-old mouse embryos [10]. The MEFs were cultured in high-glucose DMEM medium with 10% fetal bovine serum (FBS) in a 37 °C incubator with 5% CO_2_. The MEFs (3 × 10^5^, passage 1) seeded onto a 6 cm dish were cultured for three days, and the total number of cells was counted. The counted cells (3 × 10^5^) were again seeded on a 6 cm dish as passage 2. This process was repeated until passage 5.

### 2.8. Transwell Migration Assay

The migration assays were performed with 24-well Transwell chambers (6.5 mm diameter, 8 mm pore size, Corning) using *Mtbp^+/+^*, *Mtbp^+/+^*, and *Mtbp^H/−^* MEFs. The cells (5 × 10^4^) in 100 µL of 0.5% FBS-containing DMEM were seeded on the upper part of the chamber, while in the lower part of the chamber, 10% FBS-containing DMEM was added as a chemoattractant, allowing for cell migration for 14 h. The non-migrating cells were removed gently from the upper part of the chamber of the membrane using cotton swabs, while the migrating cells on the lower surface were stained with the Diff-Quik Stain Set (Dade Behring, Deerfield, IL, USA). Stained cells in the entire fields were counted using an inverted microscope.

### 2.9. Immunohistochemistry (IHC)

Tumors fixed with 10% buffered formalin for 24 h were embedded in paraffin at the Department of Pathology and Laboratory Medicine in KUMC. The tissue sections were deparaffinized in xylene and rehydrated through a series of graded alcohols. Endogenous peroxidases were inactivated with 3% hydrogen peroxide in PBS for 20 min at 25 °C. After washing with PBS, the slides were incubated in blocking solution (PBS with 0.1% Triton X-100, 3% bovine serum albumin) with 5% normal donkey serum for 10 min at 25 °C. Following antigen retrieval with sodium citrate buffer (10 mM sodium citrate, pH 6.0) for 20 min, IHC was performed using the Vector R.T.U. Vectastain Kit (PK-7800, Vector Laboratories, Newark, CA, USA). The sections were incubated with primary antibodies overnight at 4 °C and subsequently with biotinylated secondary antibodies at room temperature for 30 min. The Vector ImmPact DAB Peroxidase Substrate Kit (SK-4105, Vector Laboratories, Newark CA, USA,) was used for color development followed by hematoxylin counterstaining. All the tumors were pathologically examined.

### 2.10. Statistical Analysis

The experimental results of the RT-PCR and migration assays were analyzed using the Student’s *t*-test. The statistical analyses for immunofluorescence and IHC were performed using the two-tailed Fisher’s exact test. The statistical analysis of the survival curves was performed using the log-rank test. A *p*-value of 0.05 or lower was considered statistically significant. The statistical analyses were performed with the Graph Pad Prism software Version 9.4.1 (San Diego, CA, USA). 

## 3. Results

### 3.1. Generation of Mtbp Hypomorphic (Mtbp^H/−^) Mice

Complete deletion of *Mtbp* in mice results in early embryonic lethality, making the study of the in vivo physiological functions of Mtbp challenging [10]. To overcome this, we generated mice that carried the *Mtbp* hypomorphic (*Mtbp^H^*) allele by inserting the *hygromycin* (*hyg*) cassette surrounded by the loxP and Frt sequences into the intron 5 [21,25] (Figure 1a). When the *hyg* cassette is inserted into an intron of a gene in the same transcriptional direction, the HSV-tk polyadenylation signal in the *hyg* cassette (TKhygpA) causes early attenuation of gene transcription from the endogenous promoter [21,25]. This strategy results in only 20~30% expression of the wild-type allele in most cases [21,24,25,26]. Embryonic stem (ES) cells with the *Mtbp^H^* allele were selected with Southern blotting (Figure 1a) followed by the generation of chimeric mice. The successful germline transmission of the targeted *Mtbp^H^* allele was confirmed with genomic PCR (Figure 1a).

We additionally generated mice with a *pGKneo-SV40 poly(A)* cassette surrounded by the loxP and Frt sequences into the intron 5 as well as a loxP site in the intron 7 (Appendix A). However, the loxP site in the intron 7 was not incorporated into the genome of the mouse ES cells. When we analyzed mice with the targeted allele, we never obtained mice homozygous for the *pGKneo-SV40 poly(A)* cassette at 3 weeks of age. This observation indicates that the *Mtbp* allele with the *pGKneo-SV40 poly(A)* cassette is a null allele (*Mtbp^−^*), consistent with a previous report about embryonic lethality of *Mtbp* knockout mice [10]. Hence, the *Mtbp* allele with the *pGKneo-SV40 poly(A)* cassette was hereafter named as a *Mtbp^−^* allele. These genetically engineered mice were backcrossed to C57BL/6 for eight generations followed by the generation of *Mtbp^H/−^* mice by crossing *Mtbp^H/+^* mice with *Mtbp^+/−^* mice. The *Mtbp^H/−^* mice were viable and fertile and did not exhibit any obvious phenotypes, including cancer-prone phenotypes.

To investigate the expression of Mtbp in the *Mtbp^H/−^* mice, qRT-PCR was performed using mRNA from mouse livers taken from the *Mtbp^+/+^* and *Mtbp^H/−^* mice (Figure 1b). The level of *Mtbp* mRNA in the *Mtbp^H/−^* liver was approximately 30% of that in the *Mtbp^+/+^* liver. Consistently, the Mtbp protein level in the *Mtbp^H/−^* liver, detected with western blotting and immunohistochemistry (IHC), was 20~30% of that in the *Mtbp^+/+^* liver (Figure 1c,d). These data demonstrate the successful generation of *Mtbp* hypomorphic mice.

### 3.2. Increased Cell Migration of Mtbp^H/−^ MEFs

To further validate the phenotypes of hypomorphic Mtbp, we generated MEFs from the *Mtbp^+/+^*, *Mtbp^+/−^*, and *Mtbp^H/−^* mice. *Mtbp* mRNA expression in the MEFs was confirmed with qRT-PCR. Consistent with the results in the liver, the *Mtbp* mRNA levels in the *Mtbp*^+/−^ and *Mtbp^H/−^* MEFs were ~50% and ~30% of the *Mtbp*^+/+^ MEFs, respectively (Figure 2a), further confirming that the *Mtbp^−^* allele with the *pGKneo-SV40 poly(A)* cassette functions was a null allele, while the *Mtbp^H/−^* allele was a hypomorphic allele. To examine whether reduced Mtbp expression could contribute to immortalization of MEFs, we performed 3T3 assays using *Mtbp^+/+^* and *Mtbp^H/−^* MEFs. The reduced Mtbp levels did not contribute to cellular immortalization of the *Mtbp^H/−^* MEFs (Figure 2b), suggesting that Mtbp does not function as a typical tumor suppressor. Given the known metastasis-suppressive function of Mtbp [10,12], these MEFs were examined for their migratory potential using transwell migration assays. The migratory potential of the MEFs was negatively correlated with the levels of Mtbp, although the *Mtbp*^+/−^ and *Mtbp^H/−^* MEFs had similar migratory potential (Figure 2c). Moreover, the *Mtbp^H/−^* MEFs had an increase in nuclear phosphorylated Erk (p-Erk) in the absence of EGF stimulation as compared with the *Mtbp*^+/+^ MEFs (Figure 2d). These observations are corroborated by previous reports where MTBP inhibits cellular migration by inhibiting nuclear translocation of p-Erk, while Mtbp haploinsufficiency enhances cancer cell migration and metastasis [10,12].

### 3.3. Reduced Mtbp Enhances Liver Carcinogenesis

To address the in vivo role of Mtbp in liver carcinogenesis, we used the diethylnitrosamine (DEN)-induced liver carcinogenesis model [27]. Two-week-old *Mtbp^+/+^*, *Mtbp^+/−^*, and *Mtbp^H/−^* mice were intraperitoneally injected with DEN (25 mg/kg body weight) and were observed for tumor development (Figure 3a). The mice were euthanized when they became moribund. Major organs, including liver, lungs, intestine, lymph nodes, kidney, spleen, and brain, were investigated for primary tumors and metastatic nodules during dissection. All DEN-injected mice developed tumors in the liver, while several mice had nodules in the lungs as well (Figure 3b). The *Mtbp^H/−^* mice became moribund significantly earlier than the *Mtbp*^+/+^ mice, supporting a tumor-suppressive role of Mtbp in liver carcinogenesis (Figure 3c). The *Mtbp*^+/−^ mice also became moribund earlier than the *Mtbp*^+/+^ mice; however, the difference was not statistically significant. When we counted tumor nodules per liver, the *Mtbp^H/−^* mice tended to have higher tumor nodules (>10 nodules per liver in 12 out of 18 mice: 66.7%) than the *Mtbp*^+/+^ mice (10 out of 21 mice: 47.6%), although the difference was not statistically significant (Figure 3d). Many liver nodules were fused and uncountable, which could be due to multifocal tumors or intrahepatic metastases.

Intriguingly, eight *Mtbp*^+/+^ mice (38.1%) developed pulmonary nodules, while twelve *Mtbp^H/−^* mice (66.7%) formed pulmonary nodules (Table 1). Since DEN can induce lung tumors in mice [28], we stained lung tumor sections for TTF-1 (thyroid transcription factor-1), a marker of lung and thyroid adenocarcinomas, to distinguish whether these tumors originated from the liver or lung. All pulmonary nodules observed in the *Mtbp^+/+^* mice turned out to be lung adenoma or adenocarcinoma, while one pulmonary nodule that developed in an *Mtbp^H/−^* mouse was confirmed as HCC metastasis (Table 1).

Moreover, IHC staining of liver and lung tumors for p-Erk revealed higher intensity and increased nuclear localization of p-Erk in the *Mtbp^H/−^* tumors when compared to the *Mtbp*^+/+^ tumors (Figure 3e). These results are consistent with our previous finding that MTBP inhibits p-Erk nuclear localization [12]. Thus, the newly generated *Mtbp* hypomorphic mice are useful to identify and validate the in vivo physiological function of Mtbp.

## 4. Discussion

In this study, we generated *Mtbp* hypomorphic mice (*Mtbp^H/−^*) that express ~30% of the Mtbp levels expressed in wild-type mice and investigated the in vivo role of Mtbp in DEN-induced liver carcinogenesis. The results from the *Mtbp^H/−^* mice and MEFs corroborated with the previously reported inhibitory roles of MTBP in HCC progression and migration using human cancer cell lines [2,9,10,11,12,13], showing the significance and usefulness of the *Mtbp^H/−^* mice. Thus, the generation of mice that carry a hypomorphic allele provides a powerful tool to overcome the embryonic lethality caused by complete deletion of a gene of interest and examine the in vivo physiological function of the encoded protein. Indeed, previous studies have generated hypomorphic mice for genes whose knockout is embryonic lethal, but their heterozygous mice showed modest phenotypes. These include *Mdm2*, *Bub1*, *BubR1*, and *Pkd1* [25,29,30,31].

In addition to the inhibitory function on cell migration and cancer metastasis, MTBP has also been proposed to play roles in origin firing for DNA replication [5,8,32], mitotic progression [7], MDM2 stabilization [33,34], regulation of Myc [35,36], and promotion of cancer progression [6,17,18]. Additionally, whole-genome sequencing in dogs has identified a variant in the *MTBP* gene associated with proinflammatory processes [37]. Recently, Grieb et al. [20] showed that *Mtbp* heterozygous mice exhibit increased levels of metabolic markers in the liver and increased longevity. It would be intriguing to examine whether Mtbp alters the metabolism of DEN in the liver by using *Mtbp^H/−^* mice, which could consequently alter the DEN-induced liver tumorigenesis. Thus, MTBP appears to have diverse functions that depend on cellular and tissue-type contexts. Clinically, several studies using human cancer patients have suggested MTBP as a potential biomarker for favorable prognosis in patients with specific cancer types, including HNSCC, gastric cancer, and esophageal squamous cell carcinoma [9,11,13,14]. On the other hand, MTBP expression appears to serve as a biomarker for poor prognosis in patients with glioblastoma and triple-negative breast cancer [6,33]. Although the *Mtbp^H/−^* mice did not show a significant increase in metastases of the DEN-induced liver tumors, we observed an increase in liver and lung nodules in the *Mtbp^H/−^* mice as compared with the *Mtbp^+/+^* mice, which may be due to intrahepatic and intrapulmonary metastases or increased incidence of multifocal tumors. In either case, the results from the *Mtbp^H/−^* mice support the tumor-suppressive role of MTBP. *Mtbp^H/−^* mice would also be a useful tool to study the in vivo role of Mtbp in the progression of other cancer types. Moreover, it would be interesting to generate mice that express a high level of Mtbp and examine whether Mtbp overexpression inhibits or promotes the progression of certain types of tumors.

Additionally, the *Mtbp^H^* allele contains Frt and loxP sequences surrounding the *TKhygpA* cassette in the intron 5, while it also contains a loxP site in the intron 6 (Figure 1a). Hence, crossing *Mtbp^H/−^* mice with mice expressing Flp recombinase is expected to make *Mtbp^+/−^* mice, while crossing *Mtbp^H/−^* mice with mice expressing Cre recombinase will generate complete *Mtbp* knockout mice. When these recombinases are expressed in a specific tissue or a specific condition, mice with the *Mtbp^H^* allele would be valuable to examine tissue-type or context-dependent physiological functions of Mtbp.

## 5. Conclusions

MTBP is involved in diverse cellular functions and cancer-related processes. However, its in vivo physiological function remains unclear. This is mainly because complete deletion of *Mtbp* in mice leads to early embryonic lethality. Our newly generated *Mtbp* hypomorphic mice and their derived cells are extremely useful to investigate the in vivo physiological function of Mtbp, including its role in cancer progression and metastasis. Indeed, the *Mtbp* hypomorphic mice show enhanced DEN-induced liver carcinogenesis with increased nuclear localization of p-Erk1/2, consistent with the previous reports.

## Figures and Tables

**Figure 1 cancers-15-04596-f001:**
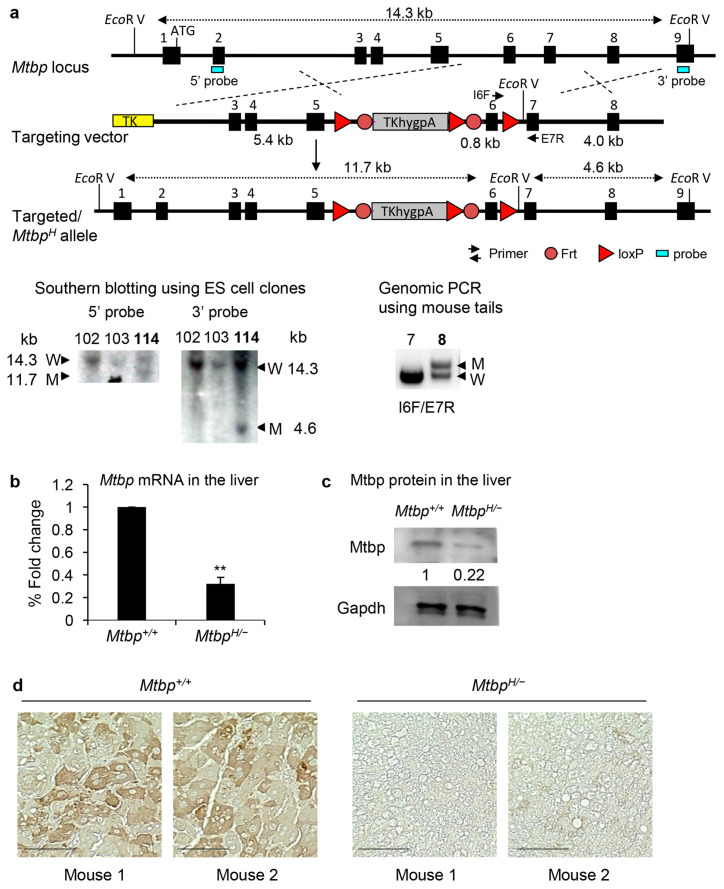
Generation and characterization of *Mtbp^H/−^* mice. (**a**) Genomic organization of the murine *Mtbp* gene, *Mtbp^H^* targeting vector, and targeted allele. The vector is constructed as a conditional allele using two different recombinase systems, Flp/Frt and Cre/loxP. The *hyg* cassette (*TKhygpA*) attenuates transcription of the endogenous *Mtbp* gene, resulting in the *Mtbp^H^* allele. Representative results of Southern blotting (**bottom**, **left**) following *Eco* RV restriction enzyme digestion of the genomic DNA from ES cell clones (#102, 103, 114) using the 5′ and 3′ probe set in exon 2 and exon 9, respectively. Genomic PCR (**bottom**, **right**) using the genomic DNA from mice (#7, 8) with primers of I6F and E7R, showing successful germline transmission. (**b**) Results of qRT-PCR for *Mtbp* using mRNA from *Mtbp^+/+^* and *Mtbp^H/−^* mouse livers. Data are normalized with values of *Gapdh* mRNA. Error bars: means + S.E. from three independent experiments. Student’s *t* test: **, *p* < 0.01. (**c**) Western blotting for Mtbp and Gapdh using protein extracts from liver tissues isolated from *Mtbp^+/+^* and *Mtbp^H/−^* mice. (**d**) IHC for Mtbp using liver tissues from *Mtbp^+/+^* and *Mtbp^H/−^* mice (2 representative images from each genotype). Scale bar, 25 μm. The uncropped blots are shown in Appendix A.

**Figure 2 cancers-15-04596-f002:**
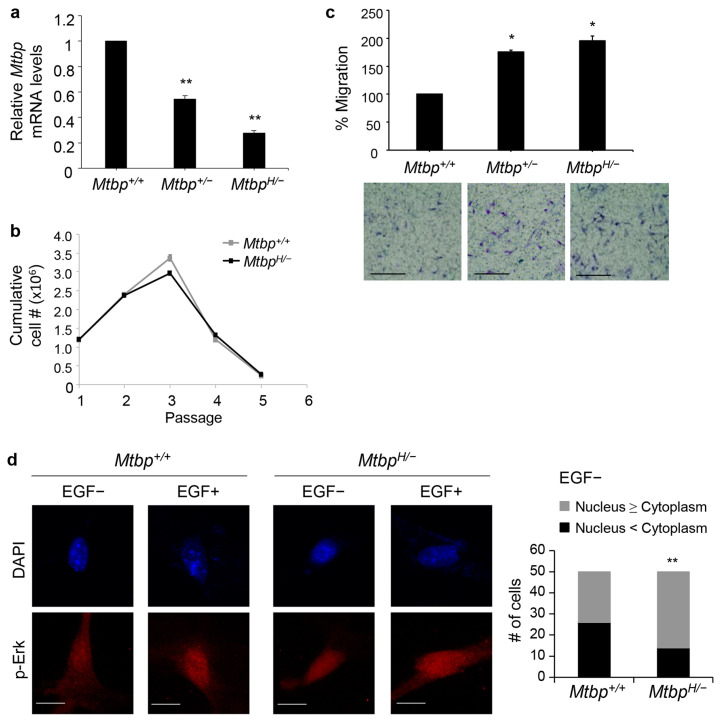
Increased cell migration of *Mtbp^H/−^* MEFs. (**a**) Results of qRT-PCR for *Mtbp* using mRNA from *Mtbp^+/+^, Mtbp^+/−^,* and *Mtbp^H/−^* MEFs. Data are normalized with values of *Gapdh* mRNA. Error bars: means ± S.E. from three independent experiments. Student’s *t* test: **, *p* < 0.01. (**b**) 3T3 assays using *Mtbp^+/+^* and *Mtbp^H/−^* MEFs. Error bars: means + S.E. from three independent experiments. Student’s *t* test: not significant. (**c**) Transwell migration assays using *Mtbp^+/+^, Mtbp^+/−^,* and *Mtbp^H/−^* MEFs. Cells were plated on the upper chambers of the Transwell. Migrating cells in the entire fields were counted 14 h later. A summary graph (**top**) and representative images (**bottom**). Scale bar, 25 μm. Error bars: means ± S.E. from three independent experiments. Student’s *t* test: *, *p* < 0.05. (**d**) Immunofluorescence studies for p-Erk following treatment with vehicle (EGF-) or 50 ng/mL of EGF (EGF+) for 30 min using *Mtbp^+/+^* and *Mtbp^H/−^* MEFs. Scale bar, 25 μm. Graph showing the number of cells with different Mtbp locations (Nucleus ≥ Cytoplasm or Nucleus < Cytoplasm) in the absence of EGF treatment (*n* = 50). Fisher’s exact test (two tailed): **, *p* < 0.01.

**Figure 3 cancers-15-04596-f003:**
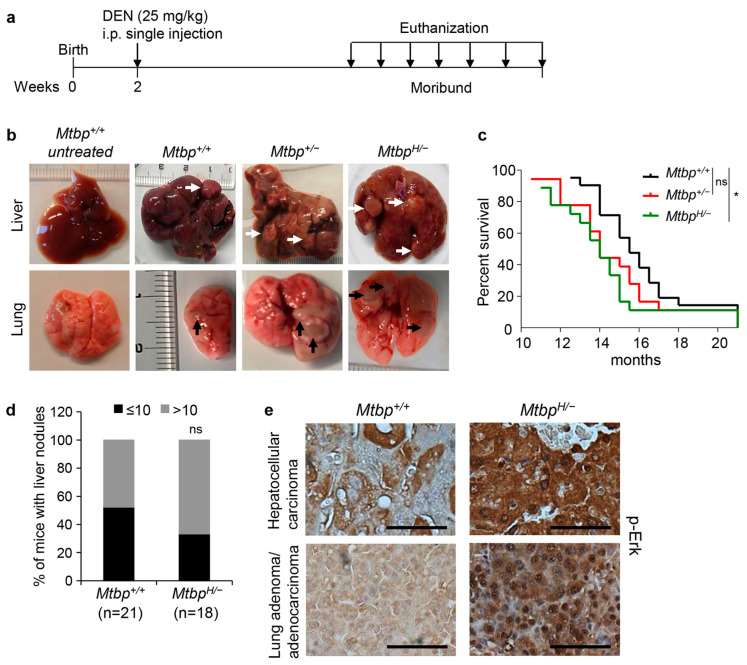
Increased liver carcinogenesis in DEN-treated *Mtbp^H/−^* mice. (**a**) Schematic of DEN-induced liver carcinogenesis studies. Mice were injected with DEN at 25 mg/kg of body weight two weeks after birth. (**b**) Representative images of the liver (**top**) and lung (**bottom**) of a non-DEN-treated *Mtbp^+/+^* mouse and DEN-treated *Mtbp^+/+^, Mtbp^+/−^,* and *Mtbp^H/−^* mice with tumors. White and Black arrows indicate tumor nodules in liver and lung, respectively. (**c**) Kaplan–Meier survival curves of *Mtbp^+/+^* (*n* = 21)*, Mtbp^+/−^
*(*n* = 18)*,* and *Mtbp^H/−^* (*n* = 18) mice injected with DEN. Log-rank test: * *p* < 0.05; ns: not significant. (**d**) Numbers of mice with liver nodules in *Mtbp*^+/+^ (*n* = 21) and *Mtbp^H/−^* (*n* = 18) mice. Fisher’s exact test (two tailed); ns: not significant. (**e**) IHC for p-Erk using HCC tumors from *Mtbp*^+/+^ and *Mtbp^H/−^* mice as well as a lung adenoma from a *Mtbp^+/+^* mouse or a lung adenocarcinoma from a *Mtbp^H/−^* mouse. Scale bar, 25 μm.

**Table 1 cancers-15-04596-t001:** DEN-induced lung tumors and metastases from the liver in *Mtbp^+/+^* and *Mtbp^H/^^−^* mice.

Genotype	Gender (Male: M, Female: F)	# of Mice	# of Mice with Lung Nodules	# of Mice with Metastatic Nodules
*Mtbp^+/+^*	M	12	5	0
	F	9	3	0
	Total	21	8 (38.1%)	0
*Mtbp^H/−^*	M	10	8	1
	F	8	4	0
	Total	18	12 (66.7%)	1

#: number.

## Data Availability

The data can be shared up on request.

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
