# Peer review of "Characterization of an Mtbp Hypomorphic Allele in a Diethylnitrosamine-Induced Liver Carcinogenesis Model"

_cancers, 2023, doi:10.3390/cancers15184596_

Round 1

Reviewer 1 Report

In this communication titled “Characterization of an Mtbp hypomorphic allele in a diethyl nitrosamine-indued liver carcinogenesis model”, authors have created a novel mice model carrying a hypomorphic Mtbp allele (MtbpH) to overcome early embryonic lethality in case of Mtbp KO condition. Through this novel model, authors evaluated the function of Mtbp in carcinogen-induced tumor and metastasis.

The presented study demonstrates a well-structured approach, and the experiments have been meticulously designed and executed. I have a few minor suggestions for improvement:

1.      Consider the inclusion of images depicting non DEN-treated WT mice liver and lung in Fig. 3b. This addition could serve as a valuable control for better clarity.

2.      To enhance the comprehensibility of Fig. 3, it might be beneficial to incorporate qRT-PCR/WB quantification of metastasis marker genes.

3.      Additionally, presenting the differentially expressed genes associated with tumorigenesis/metastasis in a tabular format for Mtbp+/+, Mtbp+/-, and MtbpH/- could provide the audience with more informative insights.

Author Response

We appreciate the reviewers taking time to read our manuscript and provide critical, in-depth feedback. We have edited the manuscript by addressing the comments raised by the reviewers, which greatly helped improving the quality of the manuscript. Below is our response to the reviewers’ comments.

Reviewer 1

1-1: Consider the inclusion of images depicting non-DEN-treated WT mice liver and lung in Fig. 3b. This addition could serve as a valuable control for better clarity.

Answer 1-1. Accordingly, we have incorporated the images of non-DEN-treated WT mice liver and lung in Fig. 3b.

1-2: To enhance the comprehensibility of Fig. 3, it might be beneficial to incorporate qRT-PCR/WB quantification of metastasis marker genes.

Answer 1-2. Thank you for kind suggestions to incorporate qRT-PCR/WB quantification of metastasis marker genes. In our previously published paper in Oncotarget (2018), we showed that MTBP suppresses metastasis, at least partially, by down-modulating the Erk1/2-Elk-1 signaling pathway. Based on this finding, we chose and examined the status of phosphorylated Erk (p-Erk) in tumors developed in Mtbp+/+ and MtbpH/- (hypomorphic) mice, as a marker of HCC malignancy and a marker of in vivo Mtbp activity. Presently, there is no defined marker for HCC metastasis. Moreover, in our study, the Mtbp hypomorphic mice developed only one lung metastasis, which precluded examination of the effects of Mtbp on distant metastasis. Thus, we will need to utilize additional tumor models and quantify the expression of metastasis markers in the future.   

1-3: Additionally, presenting the differentially expressed genes associated with tumorigenesis/metastasis in a tabular format for Mtbp+/+, Mtbp+/-, and MtbpH/- could provide the audience with more informative insights.

Answer 1-3. We agree the idea of the reviewer examining differentially expressed genes (DEGs) by RNA sequencing studies among mice with different Mtbp genotypes. Such a study will require a period of 6-12 months, which is not sufficient to complete within the timeline given to this revision. Hence, this is certainly included in the future experiments we will do. Here, we focus on establishing and validating the Mtbp hypomorphic mouse model that will be useful for a variety of future studies including DEGs among different Mtbp genotypes.

Reviewer 2 Report

This group is one of the pioneering groups of MTBP pathophysiology research in vivo and in vitro. Thus, the quality of most data they presented is fine and reportable. However, this reviewer was a little concerned about the logical structure that supported the authors' findings and, consequently, missing data.

In the life sciences and biomedicine, there are many proteins whose functions have been only partially elucidated.  MTBP is one of them. Unlike hair proteins that are not directly involved in life support, the fact that MTBP-deficient mice are extremely frail or lethal demonstrates that MTBP is functionally critical for vital activities. Therefore, any experimental approaches to exploring any part of the unknown functions of protein research using knockout (or partial KO) mice would always be expected to have the following three components:

(1) a testable hypothesis in a physiological or pathological sense,

(2) a scientifically careful experimental system based on the above hypotheses and

(3) careful in vivo and in vitro data interpretations without false assumptions, prejudices, and arbitrary expectations.

With this respect, the first thing that was difficult for this reviewer to understand was the logical justification for using liver cancer-inducing agents. Indeed, the expression of murine Mtbp is reduced by nearly 60-70% in H/− murine livers in Figures 1b, c, and d. On the other hand, the artificial transgenic strategy displayed in Figure 1a was a systemic approach and did not specifically manipulate only the liver Mtbp gene. Therefore, Mtbp protein should be predominantly decreased in all organs, including the liver. Although this reviewer is not an expert in hepatology, the most important physiological function of the liver is toxicant metabolism. In this regard, if there is already a report that MTBP is involved in hepatotoxicant metabolism, that paper should be cited to justify why the liver was chosen from among the many organs. An attractive hypothesis would then naturally be conceived to clarify the physiological role of Mtbp in the liver function of interest.

This group already reported the inhibitory role of MTBP in cell migration by interacting with and inhibiting ACTN4 function nearly 10 years ago (Oncogene. 2013 Jan 24; 32(4): 462–470). This reviewer does not understand why the current authors did not investigate the expression and function of ACTN4 not only in MtbpH/- MEF cells but also in MtbpH/- liver tissues as far as they were consistently interested in the inhibitory role of Mtbp in cell migration (see Fig. 2c). This group must have maintained a unique MtbpH/- mouse, so obtaining those normal liver (Mtbp+/+, MtbpH/-) and chemical hepatocellular carcinoma (Mtbp+/+, MtbpH/-) tissues should not be a problem.

 Minor concern: The scale bar of Fig. 3e is correct? Mouse HCC nucleus size is so small? Please correct the scale bar there.

Author Response

We appreciate the reviewers taking time to read our manuscript and provide critical, in-depth feedback. We have edited the manuscript by addressing the comments raised by the reviewers, which greatly helped improving the quality of the manuscript. Below is our response to the reviewers’ comments.

2-1: This group is one of the pioneering groups of MTBP pathophysiology research in vivo and in vitro. Thus, the quality of most data they presented is fine and reportable. However, this reviewer was a little concerned about the logical structure that supported the authors' findings and, consequently, missing data.

In the life sciences and biomedicine, there are many proteins whose functions have been only partially elucidated.  MTBP is one of them. Unlike hair proteins that are not directly involved in life support, the fact that MTBP-deficient mice are extremely frail or lethal demonstrates that MTBP is functionally critical for vital activities. Therefore, any experimental approaches to exploring any part of the unknown functions of protein research using knockout (or partial KO) mice would always be expected to have the following three components:

(1) a testable hypothesis in a physiological or pathological sense,

(2) a scientifically careful experimental system based on the above hypotheses and

(3) careful in vivo and in vitro data interpretations without false assumptions, prejudices, and arbitrary expectations.

With this respect, the first thing that was difficult for this reviewer to understand was the logical justification for using liver cancer-inducing agents. Indeed, the expression of murine Mtbp is reduced by nearly 60-70% in H/− murine livers in Figures 1b, c, and d. On the other hand, the artificial transgenic strategy displayed in Figure 1a was a systemic approach and did not specifically manipulate only the liver Mtbp gene. Therefore, Mtbp protein should be predominantly decreased in all organs, including the liver. Although this reviewer is not an expert in hepatology, the most important physiological function of the liver is toxicant metabolism. In this regard, if there is already a report that MTBP is involved in hepatotoxicant metabolism, that paper should be cited to justify why the liver was chosen from among the many organs. An attractive hypothesis would then naturally be conceived to clarify the physiological role of Mtbp in the liver function of interest.

Answer 2-1. We agree with the reviewer’s points. Indeed, the reason why we chose the liver carcinogen (DEN)-induced liver tumor model is based on observations from our previous: 1. MTBP levels are reduced in human HCC, 2. MTBP inhibits migration and metastasis of human liver cancer cells by inhibiting nuclear localization of phosphorylated Erk1/2 (p-Erk1/2). In this manuscript, we focus on characterizing and validating this unique mouse model.

We hypothesized that MTBP plays a role in inhibiting progression of liver cancer. To test this hypothesis using in vivo mouse models, we utilized our newly generated Mtbp hypomorphic mice that express only ~30% of Mtbp levels in all tissues as compared with Mtbp levels in wild type mice. We applied this mouse model to a well-established DEN-induced liver carcinogenesis mouse model, instead of generating liver-specific Mtbp conditional knockout mice. Following treatment of control wild-type and the Mtbp hypomorphic mice with a liver-specifc carcinogen, DEN, we successfully demonstrate that Mtbp hypomorphic mice show accelerated liver tumor progression.

            As the reviewer indicates, Grieb et al (Aging, 2016) show that Mtbp alters mRNA expression of several metabolic markers in the liver. Unfortunately, they did not address whether Mtbp alters the toxicant metabolism and detoxification activity in the liver. Nonetheless, it is intriguing to examine the effect of Mtbp on DEN metabolism because Mtbp could alter the metabolism of DEN as a carcinogen which could alter DEN-induced liver cancer progression. This question, as well as whether Mtbp alters physiological liver functions, should be examined in a future study, which is now described in the discussion section with citation of this paper by Grieb.

2-2: This group already reported the inhibitory role of MTBP in cell migration by interacting with and inhibiting ACTN4 function nearly 10 years ago (Oncogene. 2013 Jan 24; 32(4): 462–470). This reviewer does not understand why the current authors did not investigate the expression and function of ACTN4 not only in MtbpH/- MEF cells but also in MtbpH/- liver tissues as far as they were consistently interested in the inhibitory role of Mtbp in cell migration (see Fig. 2c). This group must have maintained a unique MtbpH/- mouse, so obtaining those normal liver (Mtbp+/+, MtbpH/-) and chemical hepatocellular carcinoma (Mtbp+/+, MtbpH/-) tissues should not be a problem.

Answer 2-2. This is a valid concern. First, in our Oncogene (2013) paper, we use only osteosarcoma cell lines to show that human MTBP inhibits osteosarcoma metastasis at least partially by inhibiting ACTN4 activity. The mechanism of metastatic progression in liver cancer is likely different from that in osteosarcoma. Second, DEN is known to activate ERK1/2 signaling. Hence, in this study, we focus on the MTBP-Erk1/2 axis following DEN administration to mice. However, as the reviewer points out, we do not exclude the possibility that DEN-induced liver cancer progression is also impacted by ACTN4 activity which could be altered by Mtbp. Since the timeline given to this revision is not sufficient to conduct different functional assays associated with ACTN4, this specific question will be addressed in the future.  

2-3: Minor concern: The scale bar of Fig. 3e is correct? Mouse HCC nucleus size is so small? Please correct the scale bar there.

Answer 2-3. We appreciate the reviewer’s careful observation. We have double-checked the size of scale bars. The used scale bar in Fig. 3E is 25 µm. The reported size of the mouse hepatocyte nucleus is between 1 µm and 9 µm. Hence, we believe that the nuclear size in the panel is within the accepted range, and the size of scale bar is correct.   

Reviewer 3 Report

The Communication article entitled “Characterization of an Mtbp hypomorphic allele in a diethylnitrosamine-induced liver carcinogenesis model” by Ranjan and colleagues reports the establishment of mice models carrying a genetically engineered hypomorphic allele of Mtbp gene, and the effect of the hypomorphic Mtbp in diethylnitrosamine-induced liver carcinogenesis.

The MTBP gene encodes the Mdm2 binding protein, a multifunctional protein that has been implicated in several critical cellular processes. It was initially identified as a binding partner of Mdm2, a well-known negative regulator of the tumor suppressor p53. MTBP participates in multiple cellular processes, including cell cycle regulation, DNA damage response, and apoptosis. Its interaction with Mdm2 can modulate the p53 pathway, affecting cell cycle arrest and apoptosis in response to DNA damage. Altered MTBP expression has been observed in various cancer types, indicating its potential role in oncogenesis. MTBP overexpression can contribute to cell survival and proliferation, potentially bypassing p53-mediated growth inhibition. Conversely, loss of MTBP function may lead to genomic instability and increased susceptibility to DNA damage, promoting carcinogenesis. Therefore, MTBP has been suggested as a potential tumor suppressor, however, the exact molecular function of MRBP in development and carcinogenesis is still ambiguous. MRBP homozygous deletion is embryonic lethal, making the biology of MRBP challenging to study. 

In this study, the author generated the hypomorphic Mtbp allele mice models by inserting the hygromycin cassette into the intron 5 of the Mtbp gene. The result was approximately 70% lower mRNA expression of Mtbp in the mice than in those with wild-type alleles. The hypomorphic Mtbp mice showed increased diethylnitrosamine-induced liver carcinogenesis and worse survival rates that were consistent with the lower expression of Mtbp in the mice's liver. The author also established an MEF cell line from the hypomorphic Mtbp mice for in vitro assays. The hypomorphic Mtbp MEF cells also showed 70% lower Mtbp mRNA expression which is consistent with the expression levels in the mice liver tissue. The cells also showed higher migration capacity and higher nuclear localization of phosphorylated Erk protein compared to wild-type cells.

In my opinion, this study is essential and particularly useful for future studies on MRBP biology in development and carcinogenesis. The study scientifically sounds. The results are clear and well-presented. The manuscript is well-written and easy to read and follow. I didn’t notice a typo or grammar error.

Author Response

The Communication article entitled “Characterization of an Mtbp hypomorphic allele in a diethylnitrosamine-induced liver carcinogenesis model” by Ranjan and colleagues reports the establishment of mice models carrying a genetically engineered hypomorphic allele of Mtbp gene, and the effect of the hypomorphic Mtbp in diethylnitrosamine-induced liver carcinogenesis.

The MTBP gene encodes the Mdm2 binding protein, a multifunctional protein that has been implicated in several critical cellular processes. It was initially identified as a binding partner of Mdm2, a well-known negative regulator of the tumor suppressor p53. MTBP participates in multiple cellular processes, including cell cycle regulation, DNA damage response, and apoptosis. Its interaction with Mdm2 can modulate the p53 pathway, affecting cell cycle arrest and apoptosis in response to DNA damage. Altered MTBP expression has been observed in various cancer types, indicating its potential role in oncogenesis. MTBP overexpression can contribute to cell survival and proliferation, potentially bypassing p53-mediated growth inhibition. Conversely, loss of MTBP function may lead to genomic instability and increased susceptibility to DNA damage, promoting carcinogenesis. Therefore, MTBP has been suggested as a potential tumor suppressor, however, the exact molecular function of MRBP in development and carcinogenesis is still ambiguous. MRBP homozygous deletion is embryonic lethal, making the biology of MRBP challenging to study. 

In this study, the author generated the hypomorphic Mtbp allele mice models by inserting the hygromycin cassette into the intron 5 of the Mtbp gene. The result was approximately 70% lower mRNA expression of Mtbp in the mice than in those with wild-type alleles. The hypomorphic Mtbp mice showed increased diethylnitrosamine-induced liver carcinogenesis and worse survival rates that were consistent with the lower expression of Mtbp in the mice's liver. The author also established an MEF cell line from the hypomorphic Mtbp mice for in vitro assays. The hypomorphic Mtbp MEF cells also showed 70% lower Mtbp mRNA expression which is consistent with the expression levels in the mice liver tissue. The cells also showed higher migration capacity and higher nuclear localization of phosphorylated Erk protein compared to wild-type cells.

In my opinion, this study is essential and particularly useful for future studies on MRBP biology in development and carcinogenesis. The study scientifically sounds. The results are clear and well-presented. The manuscript is well-written and easy to read and follow. I didn’t notice a typo or grammar error.

Answers 3-1. We highly appreciate the reviewer’s favorable comments.

Round 2

Reviewer 2 Report

The revised edition is conscientious and serious. I can't find any particular problems.